# Synergy of Active and Passive Remote Sensing Data for Effective Mapping of Oil Palm Plantation in Malaysia

**Nazarin Ezzaty Mohd Najib** [1], **Kasturi Devi Kanniah** [1,*], **Arthur P. Cracknell** [2] and **Le Yu** [3]

[1] Faculty of Built Environment and Surveying, Universiti Teknologi Malaysia, UTM Skudai 81310, Malaysia; nezzaty3@graduate.utm.my

[2] Division of Electronic Engineering and Physics, University of Dundee, Nethergate, Dundee DDI 4HN, UK; apcracknell774787@yahoo.co.uk

[3] Department of Earth System Science, Tsinghua University, Beijing 100084, China; leyu@tsinghua.edu.cn

\* Correspondence: kasturi@utm.my

**Abstract:** Oil palm is recognized as a golden crop, as it produces the highest oil yield among oil seed crops. Malaysia is the world's second largest producer of palm oil; 16% of its land is planted with oil palm. To cope with the ever-increasing global demand on edible oil, additional areas of oil palm are forecast to increase globally by 12 to 19 Mha by 2050. Multisensor remote sensing plays an important role in providing relevant, timely, and accurate information that can be developed into a plantation monitoring system to optimize production and sustainability. The aim of this study was to simultaneously exploit the synthetic aperture radar ALOS PALSAR 2, a form of microwave remote sensing, in combination with visible (red) data from Landsat Thematic Mapper to obtain a holistic view of a plantation. A manipulation of the horizontal–horizontal (HH) and horizontal–vertical (HV) polarizations of ALOS PALSAR data detected oil palm trees and water bodies, while the red spectra L-band from Landsat data (optical) could effectively identify built up areas and vertical–horizontal (VH) polarization from Sentinel C-band data detected bare land. These techniques produced an oil palm area classification with overall accuracies of 98.36% and 0.78 kappa coefficient for Peninsular Malaysia. The total oil palm area in Peninsular Malaysia was estimated to be about 3.48% higher than the value reported by the Malaysian Palm Oil Board. The over estimation may be due the MPOB's statistics that do not include unregistered small holder oil palm plantations. In this study, we were able to discriminate most of the rubber areas.

**Keywords:** mapping; oil palm; ALOS PALSAR 2; Sentinel; Landsat; Malaysia

## 1. Introduction

Oil palm plantation is among the world's largest agricultural plantation and it is expected to grow even more in the future [1]. High demand for palm oil products throughout the world has led to the expansion of oil palm plantations in Malaysia [2]. Presently, 5.4 Mha or 16.4% of the total land area of Malaysia has been planted with oil palm. Malaysia is the second largest oil palm producer in the world after Indonesia. In 2017, about 16.56 Mt of palm oil was exported to other countries [3], which is an important revenue source for Malaysia.

Although oil palm cultivation has been criticized because of its severe impact on the climate system through extensive land (forest) clearing and biomass burning for cultivation, studies have shown that oil palm plantations are effective for $CO_2$ sequestration due to their capabilities to produce higher biomass [4]. The palm oil industry predominantly contributes to the production of biomass in Malaysia with 80 Mt generated in 2015 and it is further expected to increase to 100 Mt by 2020 [5].

Nevertheless, the economic potential of oil palm biomass and its contribution towards environmental sustainability are often neglected. Studies by Tan [6] and Zahari et al [7] revealed that the oil palm frond petiole contains a large amount of fermentable sugar suggesting its potential as an excellent resource for biomaterials and bioenergy production. In order to utilize the full potential of this resource, further research is needed to understand and correctly map oil palm plantations and the availability of its biomass in Malaysia. Mapping the extent of oil palm plantations is essential for understanding the amount and magnitude of deforestation, analyzing its environmental consequences, informing policy decisions, and resource planning to determine the boundaries of plantations for accurate yield and biomass estimation. Correct estimation of oil palm biomass is important to recommend the conversion of biomass into bio-butanol rather than burning it on site and causing environmental issues that lead to climate change via the release of greenhouse gasses (GHGs).

Remote sensing provides an effective tool for oil palm plantation mapping at the regional scale and can provide repetitive observations at low cost. Optical remote sensing images have been used to detect oil palm plantations as they have red, near infrared (NIR), and mid-infrared bands that are able to distinguish oil palm crowns from other land uses [8–10]. Some of the previous studies on oil palm land use classifications used optical images [9,11–13]. These studies employed various digital image classification techniques such as k-nearest neighbor, maximum likelihood classification (MLC), support vector machine (SVM), random forest (RF), etc. However, similar spectral and structural properties with other green vegetation like rubber always creates confusion in the discrimination of oil palm from other land uses. Vegetation such as forest, rubber, and oil palm are hard to differentiate in optical images [8,14–16]. Moreover, optical data will tend to underestimate the area of oil palm plantation due to cloud cover, cloud shadow, and signal saturation at high leaf area and biomass values.

As an alternative to optical techniques, radar remote sensing is useful in classifying oil palm plantations [17]. Radar remote sensing can provide data in all-weather conditions, as well as in day and nighttime. The microwave radiation from the radar also has a good canopy and trunk penetration ability and it is sensitive to geometry, i.e., oil palm structure. The advantages in penetration ability of radar data helps in delineating smooth land surfaces such as water and soil from rough surfaces (e.g., shrubs, trees) with reference to its radar wavelength [18]. Thus, the L-band was found to be good in mapping forested vegetation and oil palms as it can penetrate tree canopies and provide information of the sub-canopy structures [18]. Radar remote sensing is better than optical remote sensing as it is useful in interacting with any soil condition, has different polarization images (horizontal–horizontal (HH), horizontal–vertical (HV), vertical–horizontal (VH), and vertical–vertical (VV)), and is sensitive to surface roughness, vegetation, dielectric properties, and moisture content [19].

Different polarization images of radar data, i.e., horizontal–horizontal (HH), horizontal–vertical (HV), vertical–horizontal (VH), and vertical–vertical (VV), provide a lot of information for oil palm tree detection by using various classification techniques and decision tree algorithms [15,16,20–24]. Lee and Bretschneider [25] reported that both C- and L-bands further increased classification accuracy, which agreed with the conclusion of a similar study [23]. Some studies were conducted to map oil palm using threshold values from HH and HV data (HH-HV > 6.5 dB) [12,18,19,21–23,26–30]. However, when this threshold was implemented in Peninsular Malaysia for detecting oil palm, it was found that water, urban, soil, and forest could not be completely distinguished from oil palm. On the other hand, the study by Koh et al [16] and Cheng et al [24] could only detect mature oil palm using this threshold. Lee and Bretschneider [25] found that a combination of microwave and optical data can successfully map oil palm trees in a heterogeneous environment with an overall accuracy of 94% compared to using Landsat (84%) and PALSAR (89%) data separately. Similar outcomes were also obtained by other studies in various environments [27,31–35]). However, care should be exercised when adopting methods developed for one location and applying them to other locations as there are various factors that affect the accuracy of the classification outcome such as scale, phenology, and variations in topography and background signals. There is no single type of data that is suitable for all oil palm areas [35]. Thus, in this study, a better detection algorithm is proposed for oil palm

detection and delineation from its adjacent land cover types such as forest, buildings, bare land, water, and other agricultural plantations. Accurate delineation of oil palm allows the demarcation of boundaries and accurate estimation of oil palm area coverage [9]. When applied in a temporal analysis, it is valuable for the detection of oil palm expansion and related land activities. The aim of this study was to use a combination of radar and optical data to detect both mature and young oil palm trees in Peninsular Malaysia.

## 2. Materials and Methods

### 2.1. Study Region

Malaysia is a tropical maritime country located in Southeast Asia and consists of Peninsular Malaysia, Sabah, and Sarawak. It experiences a humid tropical climate with average monthly temperatures ranging between 24 °C and 30 °C [36]. The size of Peninsular Malaysia is 132,090 km$^2$ [37] and consists of 11 states, as shown in Figure 1. Mineral and peat soil cover in Peninsular Malaysia is suitable for oil palm plantation. The species of oil palm planted in Malaysia is mostly from the *Elaeis guineensis* Jacq. species [38]. The states of Perlis, Kedah, Penang, Perak, and Selangor (Figure 1) receive a rainfall amount between 250 mm to 400 mm per month while other states in the east coast and south of the peninsular have recorded an average rainfall ranging from 100 mm to 150 mm. This climatological factor favors the growth of oil palm trees in Malaysia.

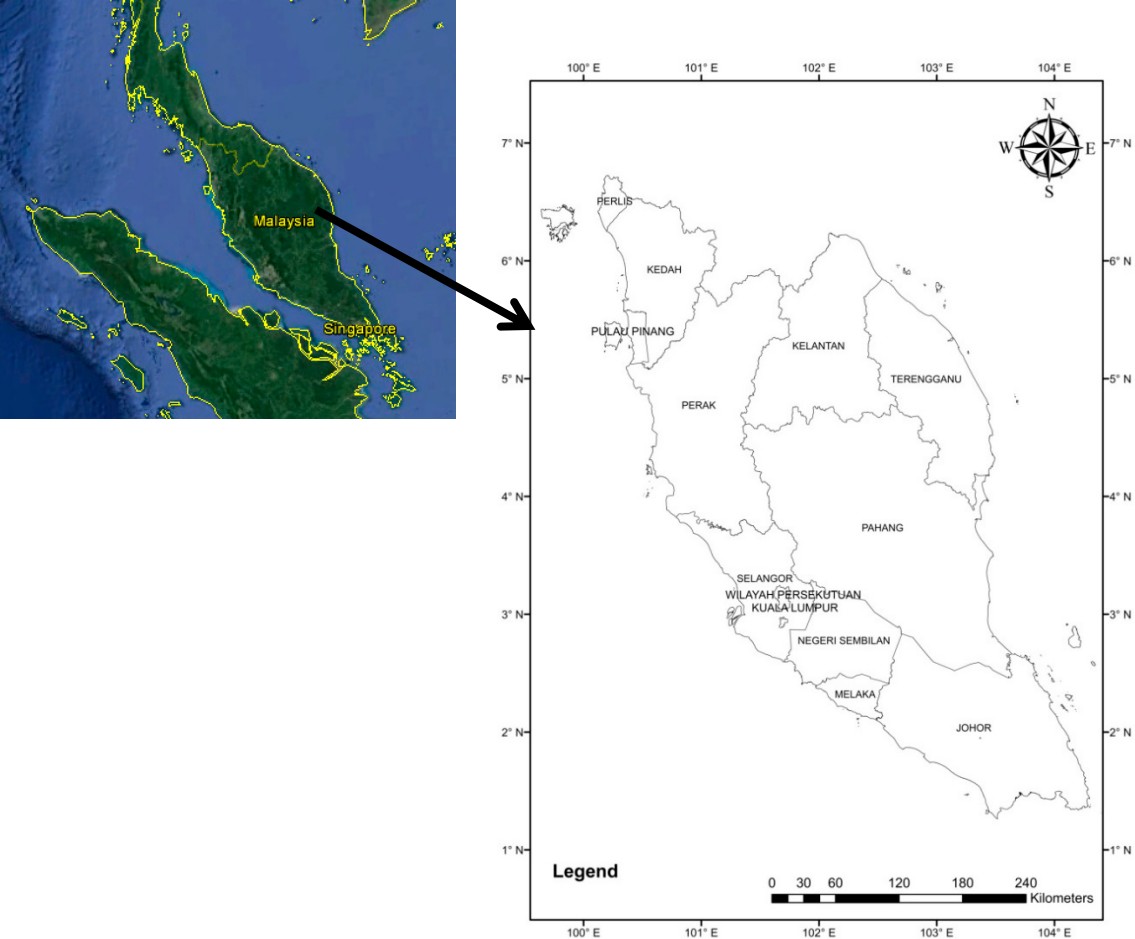

**Figure 1.** Maps showing the location of the study region. Source Figure (left): Google Earth image.

## 2.2. Datasets

The main data used in this study was 22 scenes of radar images (ALOS PALSAR 2) dated 2016. ALOS PALSAR 2 was used to detect oil palm trees and water bodies. These data with 25 m spatial resolution (JAXA MOSAIC backscattering datasets) were obtained from the Japan Aerospace Exploration Agency (JAXA) [39]. Landsat-8 images dated 2016 were also used and the 30 m spatial resolution data were downloaded from the USGS Global Visualization Viewer (GloVis) website in order to remove built up areas by using the red band [40]. About 13 Landsat scenes with <20% cloud cover were downloaded. Sentinel level 1A (2016) ground range detected (GRD) products were also obtained from the European Space Agency (ESA) [41] in order to classify bare land in the study regions. This product was projected into WGS84. The digital elevation model (DEM) from the shuttle radar topography mission (SRTM) [42] (Derek Watkins, 2016) satellite was used to detect and remove any oil palm plantation found at higher altitudes of >1000 m. A land use map produced by the Department of Agriculture, Peninsular Malaysia, dated 2015 and Google Earth images dated 2016, were used to visually assess the oil palm areas detected in the study. Furthermore, comparison of the results obtained in this study was also compared with the oil palm area statistics obtained from the Malaysian Palm Oil Board (MPOB) for 2016 [3]. The datasets and overall methods adopted to map oil palm plantations in Peninsular Malaysia are shown in Table 1 and Figure 2, respectively.

**Table 1.** Specifications of datasets used in the study.

| No. | Data | Data Description | Satellites Characteristics |
|---|---|---|---|
| 1 | ALOS PALSAR 2 | L-band Dual polarization (HH and HV) 25 m × 25 m | Orbit properties: (a) Altitude: 691.65 km (at Equator) (b) Sun-synchronous (c) Repeat cycle: 46 days (d) Sub cycle: 2 days (e) Sensor modes: Fine (40 to 70 km), ScanSAR (250 to 350 km) and Polarimetric (20 to 65 km) http://www.eorc.jaxa.jp/ALOS/en/palsar_fnf/fnf_index.htm |
| 2 | Landsat 8 OLI | Multispectral band 30 m × 30 m | Orbit properties: (a) Altitude: 705 km (near-polar orbit) (b) Sun-synchronous (c) Repeat cycle: 16 days https://glovis.usgs.gov/ |
| 3 | Sentinel | C-band Level 1A 20 m × 22 m | Orbit properties: (a) Sun-synchronous (b) Wide swath (250 km) (c) Sensor modes: Stripmap, interferometric wide swath, extra wide swath, and wave https://vertex.daac.asf.alaska.edu/ |
| 4 | Shuttle radar topography mission (SRTM) | 30 m | http://dwtkns.com/srtm30m/ |
| 5 | Malaysian Palm Oil Board (MPOB) MPOB oil Palm plantation statistic | Area of oil plantation area in 2016 | http://bepi.mpob.gov.my/index.php/en/ |
| 6 | Land use map 2015 from Agriculture Department of Peninsular Malaysia | Land use data of Peninsular Malaysia | Agriculture Department of Peninsular Malaysia |

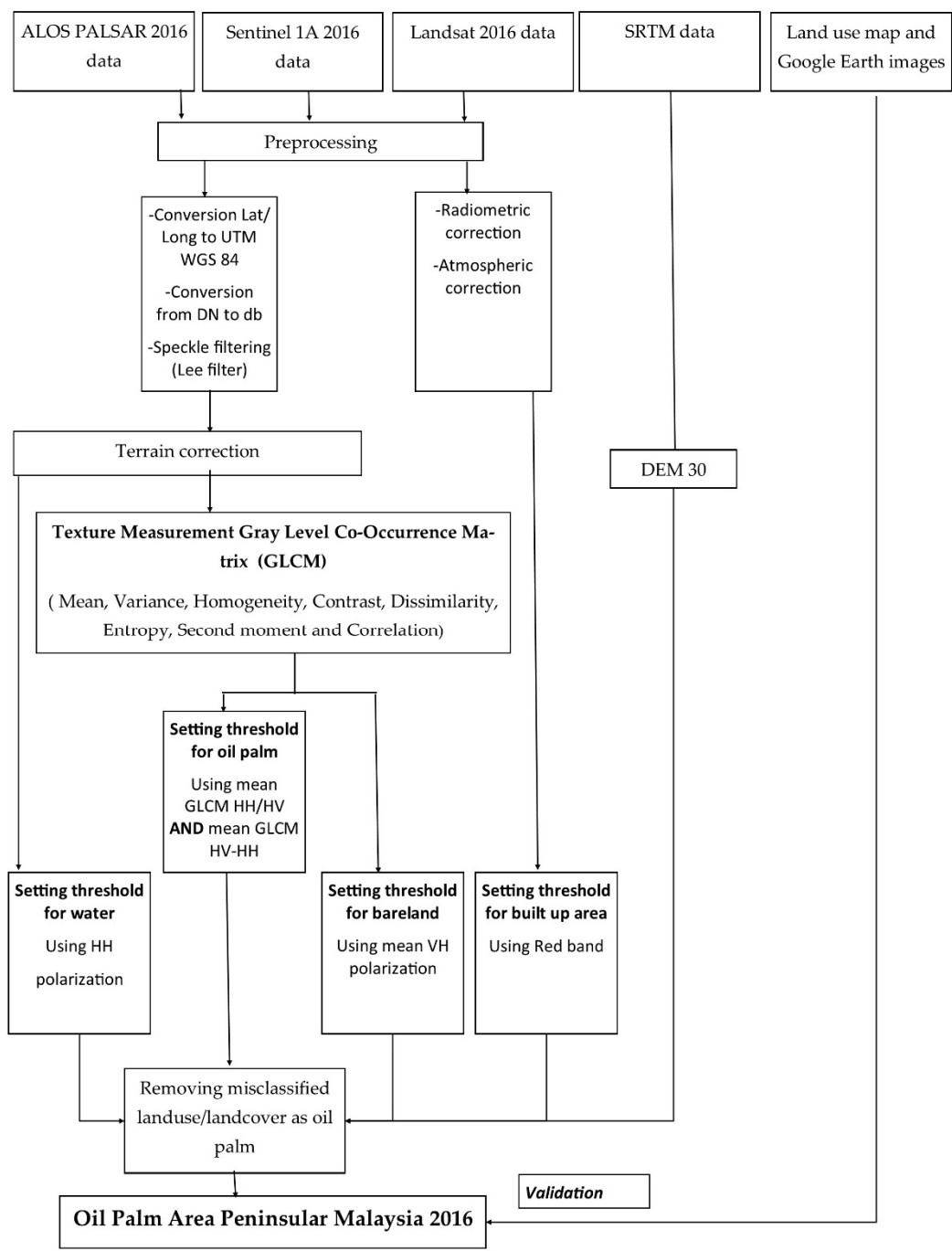

**Figure 2.** Methodology used to detect oil palm plantation in Peninsular Malaysia.

### 2.3. Methods

#### 2.3.1. Preprocessing of Radar Data

Since the ALOS PALSAR 2 L-band images provided were geometrically corrected by the agency, we only re-projected the images from the "lat long" projection system to the "World Geodetic System of 1984" (WGS 84) and radiometrically corrected and resampled the images from 25 m (original pixel size) to 30 m to match with the resolution of the Landsat images (Figure 2). The digital numbers (DN) of ALOS PALSAR 2 data were converted into $\sigma°$, dB using Equation (1) that was proposed by [43].

$$\sigma° = 10 \times \log_{10}(DN^2) + CF \qquad (1)$$

DN represents the digital number in HH and HV radar polarization. HH is radar horizontal transmission and horizontal backscattering received at the sensor while HV is radar horizontal transmission and vertical backscattering received at the sensor polarization. On the other hand, CF is the absolute calibration factor equal to −83. In order to remove noise from radar data, a Lee filter was applied using $3 \times 3$ window size. The Lee filter was used to reduce noise while preserving the sharpness, as well as to smooth the data with multiplicative components [44]. The ALOS PALSAR 2 data backscatter values were influenced by topographic effects, so terrain correction using DEM data with 30 m interval was done by the data supplier.

The C-band SENTINEL data with VH polarization was used to remove bare land areas without removing young oil palm plantations (Figure 2). The images were pre-processed for re-projecting from lat long to WGS 84 projection system, converting DN to dB, and correcting topography effects using SNAP software that has the SENTINEL level 1 toolbox [41]. The spatial resolution of the data was resampled from 10 m to 30 m to match the spatial resolution of the Landsat data. Similar to the ALOS PALSAR data the Sentinel data was also filtered for noise/speckles using the Lee filter with $3 \times 3$ window size. The speckle (Lee) filter was applied prior to texture measurement from both ALOS PALSAR and Sentinel-1 images because previous study has shown that the classification accuracy improved when speckles in the Sentinel-1 and other data were minimized using Lee filter before extracting texture information that was used in the land use/land cover [45]. However, it should be noted that when extracting texture information, the speckle can become a valuable source containing rich texture information or can cause loss of information since the function can make the image smoother. In this study, the evaluation of the error showed that without applying the Lee filter an overall classification accuracy of ~50% was obtained, but after applying the filter the accuracy increased to >70% mainly because the speckle filter reduced the salt and pepper effect from the SAR images and the texture measure (GLCM) further enhanced the backscatter signals originating particularly from oil palm plantations.

The pre-processed image was also terrain corrected using SNAP software, which provides a Range-Doppler Terrain Correction option that uses SRTM 3 sec DEM. The software determines the DEM tiles needed and downloads them automatically from the internet servers and carries out the terrain correction. The purpose of terrain correction for both ALOS PALSAR 2 and SENTINEL data is to minimize the SAR geometry effects (foreshortening, layover, and shadow) towards radar images.

### 2.3.2. Preprocessing of Optical Data

A total of 16 Landsat scenes were downloaded from the GloVis website and geo-referenced with UTM, Zone 47 North projection, and WGS 84 datum. All of the images were pre-processed using Envi 5.2 software (Figure 2). For radiometric correction the DN value of each pixel was converted into radiance using conversion factors (gains and offsets) from the providers of the satellite sensors (i.e., Landsat) [40] and then using the FLAASH function in Envi 5.2, to convert the radiance into reflectance. Clouds were manually identified and masked out using the coastal band (0.433–0.453 μm). The coastal band has been used to estimate aerosol concentration such as smoke, haze, and thin clouds in the atmosphere [46,47]. In this study, the reflectance values of the coastal band ranging between 8 and 16% could identify all types of cloud in the study region.

### 2.3.3. Texture Measurement of Radar Data

After the preprocessing, a texture measure called the gray level co-occurrence matrix (GLCM) at $3 \times 3$ window size with mean, variance, homogeneity, contrast, dissimilarity, entropy, second moment, and correlation [48] statistics was implemented to analyze the textural variation in the radar images (Figure 2). GLCM was applied to enhance the characteristics of oil palm in ALOS PALSAR data and to improve the detection of bare land in Sentinel data so that it can be removed from being misclassified as young oil palm and built up area. Subsequently, the processed ALOS PALSAR 2 images were

manipulated by finding the difference (HH minus HV) and ratio (HH over HV) between the HH and HV polarization images.

### 2.3.4. Extraction of Threshold Values

The HH, HV, HH-HV, and HH/HV images from the ALOS PALSAR, VH from Sentinel-, and red band from Landsat data were used to identify the backscattering (radar) and reflectance (optical) threshold values that can differentiate oil palm from other land covers/uses (Figure 2). This was done by studying the distributional characteristics and levels of the backscattering and reflectance values using box plots and analysis of variance (ANOVA) statistical test (Section 3.2) [22,45,49–51]. First, several ROIs (Figure 3 and Table 2) were extracted for each land cover (oil palm, forest, bare land, water, and built up areas) classes. Further we produced/plotted histograms for HH, HV, HH-HV, HH/HV, VH, and red band images using the ROIs to examine the range of backscattering and reflectance values that can represent different land covers in Peninsular Malaysia. This is rather a time consuming process that requires careful selection of pure pixels representing each land cover class in the study region. Finally, the mean and standard deviation values of each land cover class (based on the ROIs) were calculated using box plots (Figure 4). Thresholds that are specific to Peninsular Malaysia are better than the thresholds developed for oil palm areas worldwide [21] because oil palm in Peninsular Malaysia is planted in various soil surfaces (undulating) and soil types (peat and mineral soil) that may give different backscatter responses towards the satellite sensors compared to other regions.

**Table 2.** Number of samples (polygons) used for the validation of classified satellite data.

| Land Use | Oil Palm | Forest | Built Up | Bare Land | Water |
|---|---|---|---|---|---|
| No. of polygons | 1479 | 913 | 867 | 239 | 2197 |
| No. of pixels | 11,950 | 204,258 | 23,863 | 3769 | 83,908 |

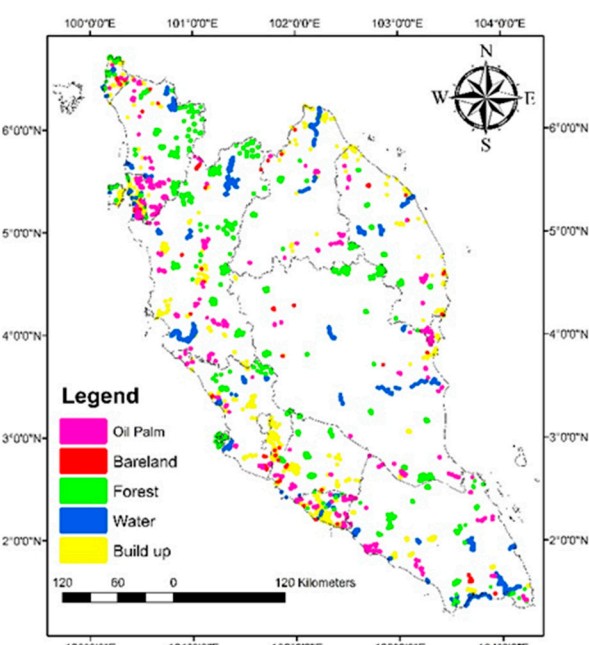

**Figure 3.** Locations of samples that were used in accuracy assessment of oil palm areas detected in Peninsular Malaysia.

The threshold values identified to discriminate oil palm in this study are generally good as they did not include most of the forest and rubber plantations as oil palm. However, sometimes the thresholds were found to include water, built up areas, and bare land. Therefore, HH polarization data was

used to set a threshold in order to remove water features (Figure 2). The HH image (after filtering for speckle) was found to effectively identify water bodies because water has low backscattering in radar images due to its smooth surface [52–55]. The HH band has been widely used for water detection such as discriminating flooded and non-flooded areas, detecting water level changes, and differentiating flood water from other water bodies [56–59].

For separating built up areas from oil palm, band 3 of Landsat data was used in this study (Figure 2). Band 3 (red) can separate built up from bare land and also young oil palm that usually contains exposed bare land. The ANOVA was computedto determine whether there were any statistically significant differences in the mean radar backscatter/reflectance values among different land cover/use classes in Peninsular Malaysia. ANOVA is useful to determine if various spectral bands/manipulation of bands of remote sensing data can differentiate oil palm from other land cover classes. Since oil palm is usually planted in areas that are lower than 1000 m elevation, any misclassification of oil palm over locations exceeding 1000 m elevation was eliminated using the DEM from the shuttle radar topography mission (STRM) satellite data (Figure 2).

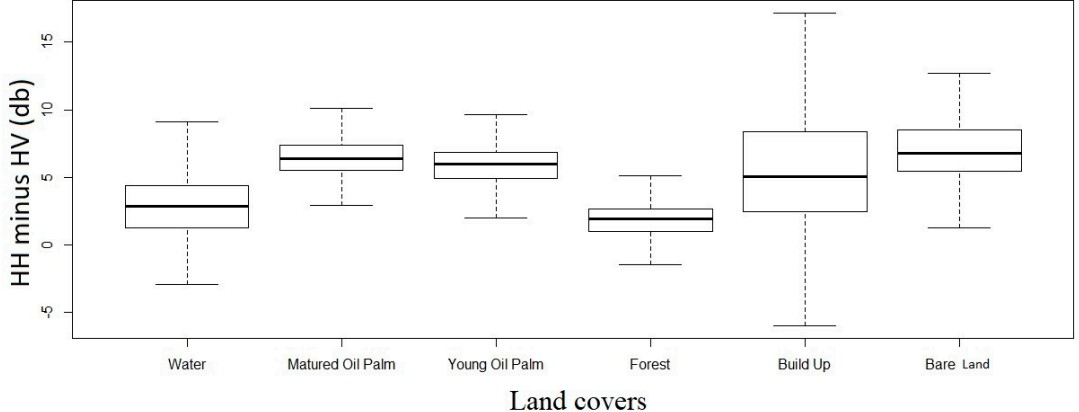

(**a**) Boxplot showing the average values of ALOS PALSAR 2 HH-horizontal–vertical (HV) for each land cover image (ANOVA-$p$ = 0, F value = 509).

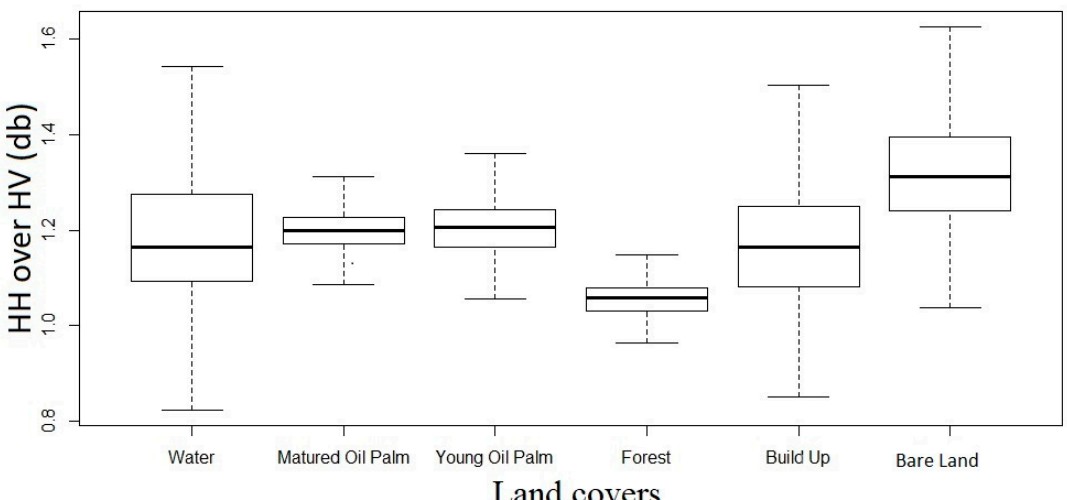

(**b**) Boxplot showing the average values of ALOS PALSAR 2 HH over HV for each land cover (ANOVA-$p$ = 0, F value = 219).

**Figure 4.** *Cont.*

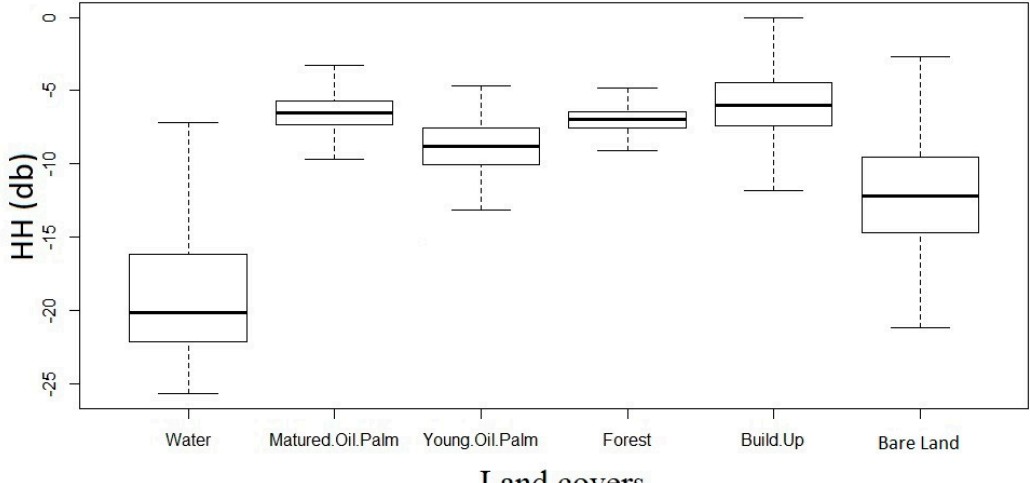

(**c**) Boxplot showing the average values of ALOS PALSAR 2 HH for each land cover (ANOVA-$p$ = 0, F value = 3400).

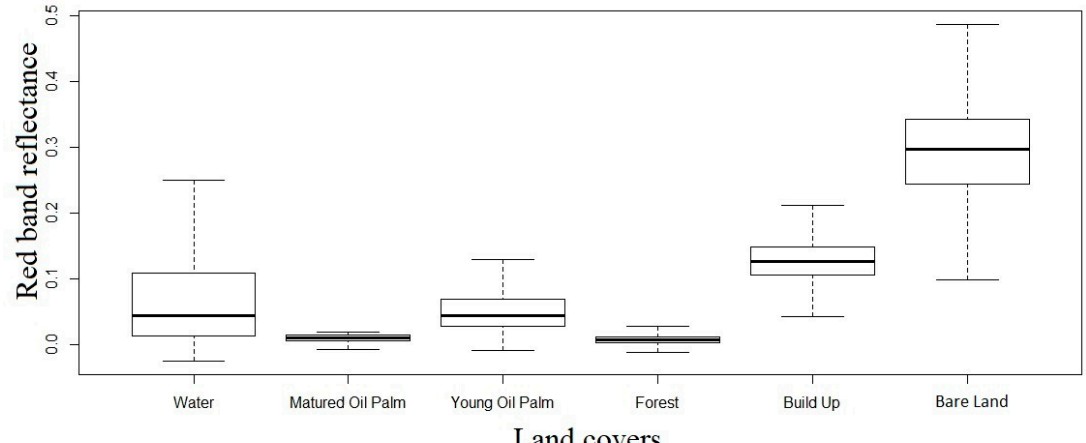

(**d**) Boxplot showing the average values of red band reflectance for each land cover (ANOVA- $p$ = 0, F value = 651).

**Figure 4.** *Cont.*

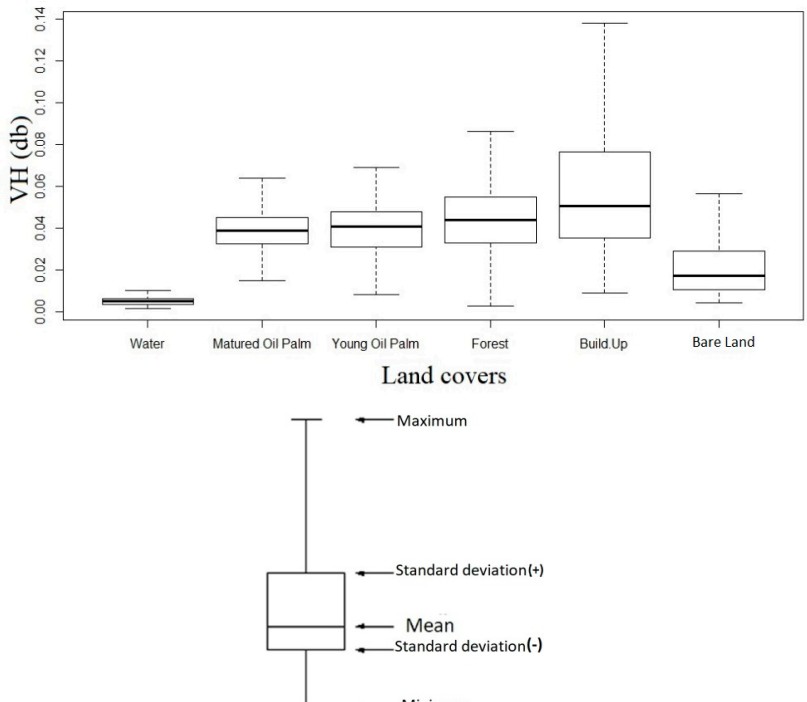

(**e**) Boxplot showing the average values of Sentinel 1 VH band for each land cover (ANOVA- *p* = 0, F value = 246).

**Figure 4.** Boxplots showing land cover type with value from satellite image.

### 2.3.5. Accuracy Assessment

The locations of oil palm detected using remotely sensed data were also verified using ground truth polygons that were extracted from Google Earth images (Figure 2). The Google Earth images were used since they provide near real time data that match the satellite images. More than 1000 polygons (regions of interest) for all land cover classes (oil palm, water, built up, bare land, and forest) with each polygon containing more than 10 pixels were digitized from Google Earth data to calculate the confusion matrix (Table 2 and Figure 3). A confusion matrix contains producer, user, overall, and kappa accuracies. The producer accuracy calculates how well a certain area has been classified (how often are real features on the ground correctly shown on the classified map). User accuracy represents how well the map shows what is really on the ground (i.e., how often the class on the map will actually be present on the ground). table In remote sensing terms, if the kappa coefficient equals 0, it means that there is no agreement between the classified image and the ground truth image. Yet if the kappa coefficient equals 1, it shows that the classified image and the ground truth image are totally identical. Therefore, the higher the kappa coefficient the more accurate the classification results are. Table 2 shows the samples used for validation of classified satellite data.

Finally, the accuracy of the oil palm areas detected in this study was assessed by comparing the results with the oil palm area statistics produced by the MPOB and the land use map produced by the Department of Agriculture, Peninsular Malaysia for all 11 states in Peninsular Malaysia.

## 3. Results

### 3.1. Detection of Oil Palm in HH and HV Polarization Radar Data

The average HH ALOS PALSAR image backscattering values for six land use/cover classes for Peninsular Malaysia are shown in Table 3. The increase and decrease (high or low) of radar signals in both HH and HV polarizations depend on the physical environment of the study area. HV polarization

is sensitive towards vegetation structure and biomass [60], while the backscatter of HH polarization from built up areas is higher than other land use/cover types due to its rough surface. [7].

**Table 3.** Average backscattering values (dB) for various land cover/land use types derived from horizontal–horizontal (HH) polarization of ALOS PALSAR radar data.

| Land Cover | | | | | |
|---|---|---|---|---|---|
| **Water** | **Matured Oil Palm** | **Young Oil Palm** | **Forest** | **Built Up** | **Bare Land** |
| −18.73 ± 4.59 | −6.53 ± 1.36 | −8.90 ± 1.76 | −7.04 ± 1.32 | −5.86 ± 2.70 | −12.10 ± 3.36 |

The extraction of land cover from HH polarization of ALOS PALSAR data shows that water can be distinguished from oil palm as the backscatter value of water tends to be lower compared to other land cover classes (water: −18.73 ± 4.59). Water appears to be black in ALOS PALSAR 2 HH radar images because it has a smooth surface (Table 3). The smooth surface of the water causes the radar signal to be directed away from the receiver and this affects the reading of the backscatter value. Water bodies recorded the lowest reading compared to other land use/covers [61]. It was found that the value for water is almost two times lower than the backscattering value of other land uses, thus it is distinguishable from oil palm. Therefore, the HH band from ALOS PALSAR 2 is suitable to separate water from other land uses especially oil palm as the backscatter does not mix with other land use types.

In order to obtain a better detection of oil palms and to minimize any misclassification of oil palm trees, polarization difference (HH minus HV) and polarization ratio (HH over HV) images of ALOS PALSAR2 radar data were integrated in this study. The radar backscatter values extracted from the polarization difference (HH minus HV) and ratio (HH over HV) images of ALOS PALSAR 2 for different land cover types found in Peninsular Malaysia are shown in Figures 4 and 5, respectively. Forest is found to provide lower backscatter values compared to oil palm in both HH minus HV polarization and HH/HV ratio images. Nevertheless, it was found that to some extent oil palm tends to be mixed with bare land, built up areas, and water bodies in both the HH-HV and HH/HV images (Figure 4; Figure 5). The analysis of variance (ANOVA) test shows a large F value of 509 and 219 for HH-HV and HH/HV images, respectively, although the p value is 0, indicating the difference in the mean radar backscattering values among the land cover classes is not large. On the other hand, the HH polarization image was able to separate water from oil palm. The average backscatter value in the HH-HV data for forest was 1.68 ± 1.67 dB. For matured and young oil palm, the backscatter values were twice as high with 6.49 ± 1.55 dB and 5.95 ± 1.79 dB, respectively. Generally, forest has lower backscatter values compared to oil palm. On the other hand, the backscatter values in the ratio image for matured and young oil palm were 1.20 ± 0.05 dB and 1.21 ± 0.07 dB, respectively, and the values are close to build up and water bodies.

### 3.2. Detection and Discrimination of Misclassified Land Covers from Oil Palm

As mentioned previously, the values of mature and young oil palm were distinguishable from forests where the radar backscatter values for oil palm was higher compared to forest in the HH -HV polarization image (Figure 5). However, oil palms especially young palms still tended to be confused with bare land and built up areas. This is due to the small size of an oil palm plantation canopy during the young age. Young palm oil trees have an under-developed canopy exposed to bare soil. When looking at a satellite pixel of 30 m, the soil that surrounds a young oil palm plantation will be dominant and thus most of the young oil palm has been misclassified as bare soil. Sentinel C-band data with VH polarization is useful to detect bare land and discriminate it from young palm oil trees. The backscatter values of bare land in Peninsular Malaysia were found to range between 0.0043 dB and 0.1896 dB (Figure 4e). Although a young palm oil tree canopy is sparse and the surrounding soil can be dominant in giving reflection towards the satellite, Sentinel C-band radar data was able to differentiate between pure bare land and bare land that has a little vegetation on it like young oil palm, due to its

canopy penetration ability [6]. Bare land appears dark in radar images because of the surface having rough and dry soil properties that cause the radar signal to be scattered away from the sensor. Dry soil allows some of the radar energy to penetrate into the soil surface while wet soil allows electrical properties between air and water to produce higher radar backscatter. The interaction between soil moisture and Sentinel 1 data studied in North Africa [57] shows a high correlation between C-band backscatter and soil in areas with low vegetation density because the presence of dense vegetation can produce a disruptive effect on the radar signal. C-band is also found to be very suitable for soil surface studies as it is able to produce the most accurate results compared to L and P bands [58].

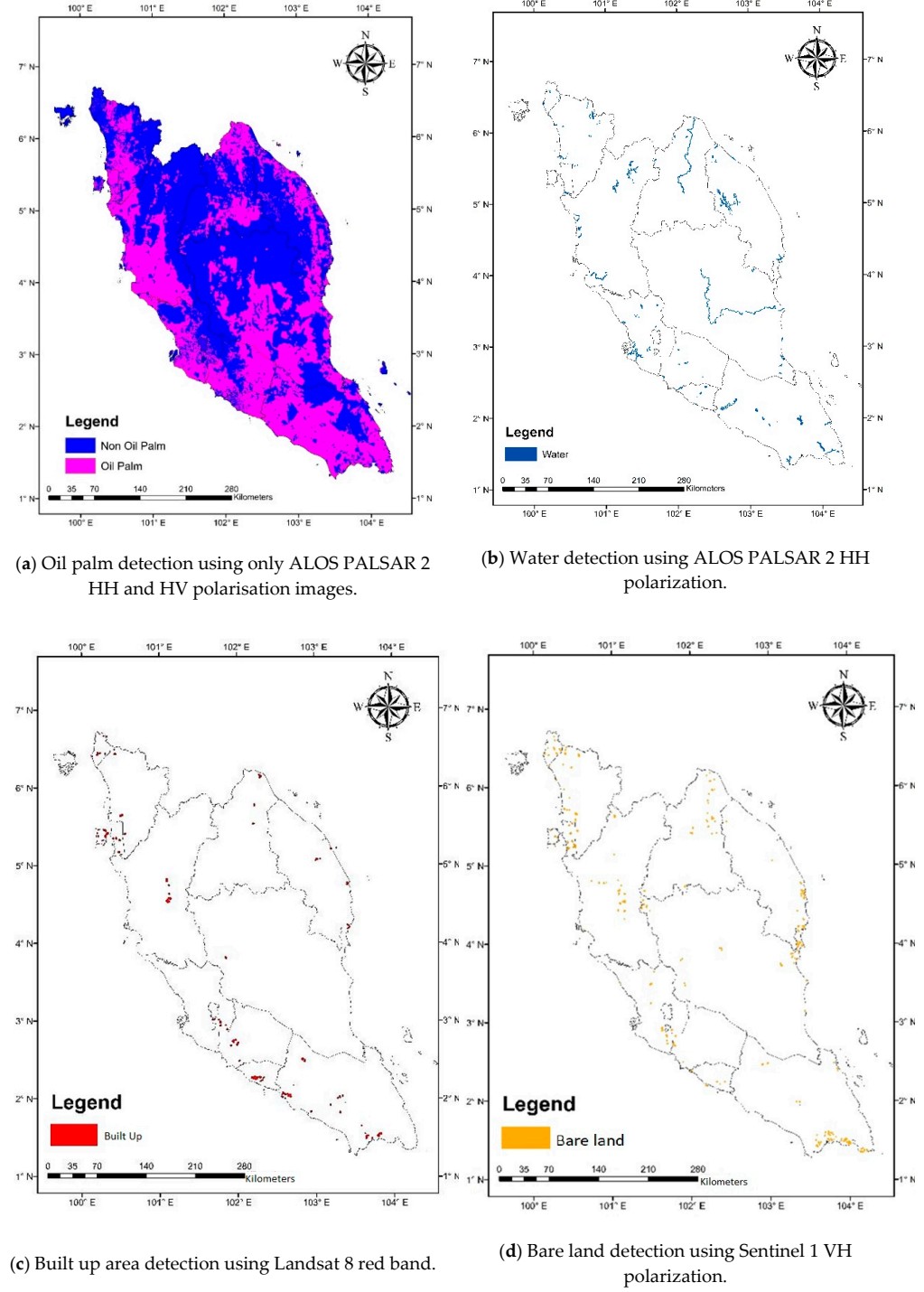

(**a**) Oil palm detection using only ALOS PALSAR 2 HH and HV polarisation images.

(**b**) Water detection using ALOS PALSAR 2 HH polarization.

(**c**) Built up area detection using Landsat 8 red band.

(**d**) Bare land detection using Sentinel 1 VH polarization.

**Figure 5.** *Cont.*

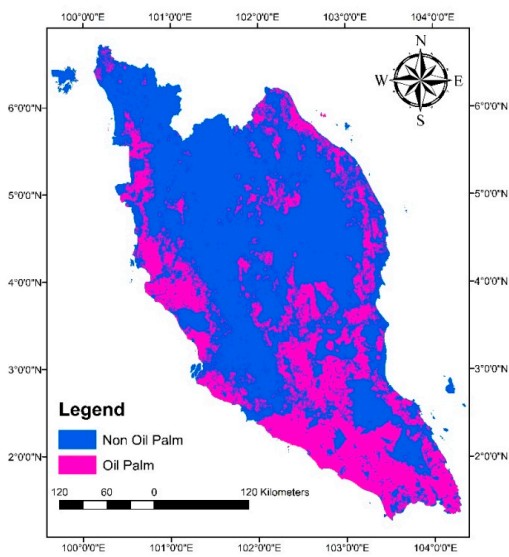

(**e**) Final map showing the areas planted with oil palm.

**Figure 5.** Oil palm areas delineation in Peninsular Malaysia using ALOS PALSAR, Sentinel 1C, and Landsat Thematic Mapper images.

The red band of Landsat 8 reflectance could detect built up areas as they recorded higher readings compared to oil palm, water, and forest but lower than bare land. There exists a significant difference among the land cover classes using the red band ($p = 0$, F value = 650.78), as shown in Figure 4d. Built up areas had higher average reflectance values of 0.11 ± 0.03 compared to the reflectance of matured oil palm (0.009 ± 0.008) (Figure 4d). Bare land showed higher reflectance values compared to built up areas because it neither consisted of any vegetation nor any other elements (e.g., building, vehicle, parks, etc.) on the Earth's surface that have low reflectivity compared to the soil of bare land. A lower reflectance was due to the absorption of the sunlight by the dense vegetation while a higher reflectance value was caused by the reflection of sunlight from a dense built up area [62]. The red band from Landsat 8 is suitable to be used to remove built up areas from oil palm. When the p value is less than 0.05 it indicates a highly significant difference in the land cover classes (Figure 4d).

### 3.3. Mapping of Oil Palm

The analysis of the HH, HV, HH-HV, and HH/HV polarization images of ALOS PALSAR 2, VH polarization image of the Sentinel 1 and the red band of Landsat TM data show that different satellite images have the potential to discriminate oil palm from other land use types when they are synergized. The backscattering and reflectance threshold values from all the datasets used in this study are shown in Table 4. Backscattering values of more than 3.5 and 1.09 dB in HH-HV and HH/HV data, respectively, are useful to identify oil palm trees throughout Peninsular Malaysia. Nevertheless, some misclassification of oil palm trees that maybe caused by bare land, water surfaces, and built up areas can be minimized by adopting radar C-band (VH) and L-band (HH) backscattering threshold values of >0.012 and <−10 dB, respectively, and reflectance (red band) of > 0.045% (Table 4 and Figure 5). By removing unwanted land use/cover types, the area of oil palm detection was improved from 4.25 Million ha (Figure 5a) to 2.77 Million ha (Figure 5f).

**Table 4.** Satellite data (optical and radar) threshold values for different land use/cover in Peninsular Malaysia.

| Land use | Oil Palm | Oil Palm | Built Up | Bare Land | Water |
|---|---|---|---|---|---|
| | (HH-HV) | (HH/HV) | (Band 3-Red Band) | (VH) | (HH) |
| Threshold | >3.5 | >1.09 | >0.045 | >0.012 | <−10 |

### 3.4. Validation of Oil Palm Mapping

The oil palm areas detected and mapped using various datasets as described in the method section were validated to determine the robustness of the techniques used in this study. We used samples shown in Figure 3 for the accuracy assessment. Table 5 shows the accuracy (percentage) of different land covers classified by applying the threshold values from HH-HV and HH/HV polarization images of ALOS PALSAR 2 data (Table 4). This algorithm could separate oil palm from forest accurately in almost 98% of cases. However, built up, bare land, and water bodies were found to mix with oil palm, thus their accuracies were much lower than forest land cover (Table 5a). After developing thresholds to identify and remove the misclassified layers from oil palm (Table 4), the accuracy of oil palm increased substantially from 83% to 98% (Table 5b). The accuracy of other land uses also increased (Table 5b). The overall confusion matrix resulted using two classes (oil palm and non-oil palm) (Table 6). It was found that not all the oil palm on the ground was presented by the classified image, however most of the classified oil palms from our classification was found correctly located on the ground truth image and this is shown by the high user and producer accuracies (Table 6).

The area of oil palm classified in this study was compared with the oil palm plantation area statistics provided by MPOB and other sources. Table 7 shows a comparison of the percentage coverage of oil palm relative to the state land area for each of the 11 states in Peninsular Malaysia. The results show that Johor had the largest coverage of oil palm in 2016 and this was followed by Melaka and Negeri Sembilan with more than 20% of the states covered by oil palm. A similar pattern was also found in the data produced by MPOB although the percentage coverage is different. Other studies that also estimated oil palm coverage in Peninsular Malaysia produced a different pattern [30]. Overall, this study estimated about 21% of Peninsular Malaysia is cultivated with oil palm, whereas MPOB reported 20.3% and the study of [30] reported much higher values of 28%.

**Table 5.** Accuracy (%) of each land cover before (**a**) and after (**b**) the misclassified land covers were removed from oil palm areas.

| (a) | | | | | |
|---|---|---|---|---|---|
| **Land Use** | **Oil Palm** | **Forest** | **Built Up** | **Bare Land** | **Water** |
| Overall accuracy (%) | 83.27 | 97.71 | 61.93 | 64.35 | 60.63 |
| (b) | | | | | |
| **Land Use** | **Oil Palm** | **Forest** | **Built Up** | **Bare Land** | **Water** |
| Overall accuracy (%) | 98.36 | 98.97 | 88.46 | 85.27 | 96.50 |

**Table 6.** Producer, User and Overall classification accuracies for oil palm and non-oil palm classes before and after removal of the misclassified pixels.

| Classification | Oil Palm | | Non-Oil Palm | | Overall Accuracy (%) | Kappa Coefficient |
|---|---|---|---|---|---|---|
| | Producer Accuracy (%) | User Accuracy (%) | Producer Accuracy (%) | User Accuracy (%) | | |
| Before removal | 80.23 | 17.45 | 83.38 | 99.11 | 83.27 | 0.21 |
| After removal | 81.79 | 76.1 | 99.01 | 99.29 | 98.36 | 0.78 |

**Table 7.** Comparison of percentage area planted with oil palm in each state in Peninsular Malaysia as estimated in this study with that of MPOB and other studies.

| State | (1) This Study | (2) MOPB (2016) | (3) Cheng et al., 2019 [30] | Difference (1) and (2) | Difference (1) and (3) |
|---|---|---|---|---|---|
| Johor | 53.67 | 38.87 | 54.73 | 14.80 | −1.06 |
| Selangor | 15.08 | 17.42 | 31.37 | −2.35 | −16.29 |
| Perlis | 1.09 | 0.80 | 19.65 | 0.29 | −18.56 |
| Kedah | 8.21 | 9.27 | 20.06 | −1.05 | −11.85 |
| Perak | 17.65 | 18.96 | 25.73 | −1.31 | −8.08 |
| Kelantan | 10.44 | 10.32 | 15.27 | 0.12 | −4.83 |
| Pulau Pinang | 9.10 | 13.54 | 19.16 | −4.45 | −10.06 |
| Melaka | 50.41 | 33.85 | 42.20 | 16.56 | 8.21 |
| Pahang | 17.13 | 20.30 | 23.85 | −3.17 | −6.72 |
| Terengganu | 13.04 | 13.22 | 19.23 | −0.18 | −6.19 |
| Negeri Sembilan | 20.70 | 26.83 | 26.98 | −6.12 | −6.28 |
| Total | 20.99 | 20.27 | 27.61 | 0.72 | −6.62 |

Further, Table 8 provides the percentage difference of total oil palm areas in this study and other studies relative to the area provided by MPOB for each state in Peninsular Malaysia. The oil palm area as detected in this study is slightly (3.48% or 94945 Ha) higher than the statistics provided by MPOB (Table 8). Analysis of the difference in each state in Peninsular Malaysia shows that the largest difference in oil palm area was found in Melaka, followed by Johor. The reason for the overestimation is provided in Section 4.2.

**Table 8.** Comparison of oil palm areas estimated in this study and other studies with that of MPOB statistics. Unit of the values is percentage (%) difference calculated as: This study-MPOB)/((This study + MPOB)/2).

| State | This Study | Cheng et al., 2019 [30] | Shaharum et al., (2020) [63] |
|---|---|---|---|
| Johor | 31.99 | 33.9 | 6.5 |
| Selangor | −14.43 | 57.18 | 35.28 |
| Perlis | 30.55 | 184.34 | 91.76 |
| Kedah | −12.07 | 73.59 | 51.18 |
| Perak | −7.17 | 30.3 | −3.49 |
| Kelantan | 1.11 | 38.68 | −22.59 |
| Pulau Pinang | −39.28 | 34.36 | −3.12 |
| Melaka | 39.32 | 21.96 | −22.5 |
| Pahang | −16.94 | 16.07 | −2.84 |
| Terengganu | −1.4 | 37 | −2.61 |
| Negeri Sembilan | −25.76 | 0.58 | −0.27 |
| Total | 3.48 | 30.67 | 3.16 |

## 4. Discussion

### 4.1. Oil Palm Mapping with Radar (Backscattering) and Optical (Reflectance) Data Threshold

The range of radar backscatter values to identify oil palm areas may vary between Peninsular Malaysia and other regions. This could be due to the influence of surface roughness, dielectric properties, moisture content, polarization sensitivity/frequency, imaging possibility from different types of polarized energy (HH and HV), and volumetric analysis [61]. In Peninsular Malaysia, oil palm is also planted at elevated locations although generally it is found on terrain with <300 m of sea level. Oil palms planted on different soil types (peat versus mineral soil) that have different moisture content also can influence the radar backscatter value. Another issue that can also affect the radar backscatter values is flood in low lying plantations.

The backscatter values of bare land in Peninsular Malaysia are found to range between 0.0043 dB and 0.1896 dB (Figure 4e). Although the canopy of young palm oil trees is sparse and the surrounding

soil can be a dominant factor in giving reflection towards the satellite, Sentinel C-band radar data was able to differentiate between pure bare land and bare land that has little vegetation on it like young oil palm, due to the C-band's canopy penetration ability [64]. Other studies such as [30] produced a large difference of 31% and [63] a small difference of only 3.2% compared to MPOB's estimation (it should be noted that [63] produced oil palm area estimates for 2017). Bare lands appear dark in radar images because of the surface, having rough and dry soil properties that cause the radar signal to be scattered away from the sensor. Dry soil allows some of the radar energy to penetrate into the soil surface while wet soil allows electrical properties between air and water to produce higher radar backscatter. The interaction between soil moisture and Sentinel 1 data was studied in North Africa [61] and they concluded that the correlation between C-band backscatter and soil can be high in areas with low vegetation density because the presence of dense vegetation can produce a disruptive effect on the radar signal. C-band is also found to be suitable for soil surface studies as it is able to produce the most accurate results compared to L and P bands data [64].

The threshold values from ALOS PALSAR 2 data as computed in this study (Table 4) are different from previous studies that map oil palm using backscattering values from HH and HV data (HH-HV > 6.5 dB) [17,22–24,28–31]. This is because the polarization combination (HH and HV) tends to give different values according to the condition of the oil palm trees (planting styles) and surrounding environmental conditions. Each geographical location may have different planting styles, topographical condition, soil type, and environment that affect the backscatter values of oil palm trees. As mentioned previously, HH polarization images of ALOS PALSAR data produced a higher backscattering signal for built up areas compared to other land use/cover types due to its rough surface [59]. Thus, the range of radar backscatter in those areas with negative values might be due to the effects from the built up areas.

Previous studies such as that of [23] used ALOS PALSAR L-band satellite data to estimate the global coverage of oil palm trees. We conducted an error analysis by implementing a previously proposed algorithm [23] for Peninsular Malaysia and the results showed that the threshold values for HH-HV and HH/HV polarizations were not able to differentiate oil palm from water, forest, and urban areas for Peninsular Malaysia. A recent study [63] projected the total oil palm areas in Peninsular Malaysia to be 21.14 Mha for year 2017. They employed non-parametric machine learning algorithms such as SVM, CART, and RF using Landsat satellite images on the Google Earth Engine platform. Although the SVM technique recorded the best overall classification accuracy with 93.16%, RF (overall accuracy of 80.1%) showed the best extractions of oil palm plantation when compared to the estimates by MPOB (area percentage difference of 3.16%). When comparing the results obtained in this study with another study's [31], our estimates are close to the estimates provided by MPOB. Cheng et al. [29] used ALOS PALSAR data with MLC technique instead of the threshold technique used in the current study. As such, they [30] misclassified urban/built up areas and croplands/other palms as oil palm. Since oil palm is planted near urban and settlement areas in Peninsular Malaysia, the tendency for oil palm to be mixed with built up area is high and therefore the built up areas must be identified and removed from oil palm [65]. In rugged or mountainous regions, topographic factors such as slope and aspect can considerably affect vegetation reflectance [65].

*4.2. Misclassification of Oil Palm (Source of Errors)*

The total area of oil palm identified in this study was larger than the statistics provided by MPOB. Therefore, in this section the reasons for the overestimation are discussed. An analysis of states in Peninsular Malaysia (Table 8) shows that the largest difference was recorded in Melaka and Johor. The largest difference noted in the state of Johor specifically might be due to the large number of small holders' plantations not registered with MPOB [66]. Thus, it shows that not all oil palm estates located in Johor were included in the MPOB statistics. Since 1999, Johor has recorded the highest number of small holder plantations in Peninsular Malaysia [67].

Moreover, some of the small holders' plantations do not have an exact area calculation because they might have additional plantations on their own lands after registering with MPOB. Since in this study we used optical imagery to remove the built up areas, there may be a limitation of removal due to cloud cover. Figure 6 shows one of the locations (1°21′23.45″ N, 103°33′49.05″ E) where cloud cover from a Landsat 8 image (Figure 6c) caused misclassification (built up area was misclassified as oil palm, Figure 6a,b) and it was irremovable from the classified image. Therefore, this limitation may have also contributed to the overestimation of oil palm area.

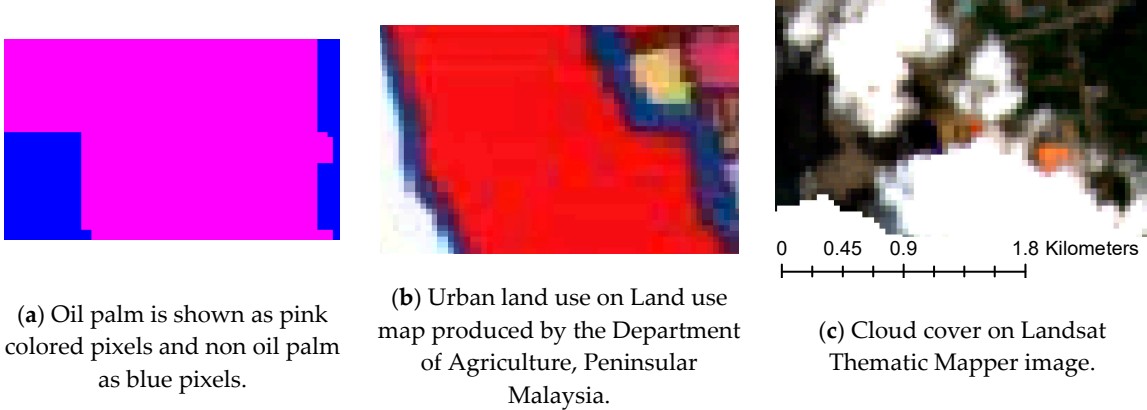

(**a**) Oil palm is shown as pink colored pixels and non oil palm as blue pixels.

(**b**) Urban land use on Land use map produced by the Department of Agriculture, Peninsular Malaysia.

(**c**) Cloud cover on Landsat Thematic Mapper image.

**Figure 6.** (**a**) Misclassified built up area as oil palm; (**b**) land use map; and (**c**) Landsat 8 image.

Another possible reason for the overestimation is the mixture of oil palm trees with coconut trees. Coconut has a similar spectral pattern (backscatter values) in ALOS PALSAR L-band data to oil palm and it should be noted that coconut trees are also planted in some of the oil palm estates in Malaysia, thus they have a similar planting style. The first author of this manuscript noticed (during her field visit) that large areas of oil palm estate owned by MPOB in Teluk Intan in the state of Perak in northern Peninsular Malaysia were planted with coconuts. Therefore, it is hard to differentiate coconut trees from oil palm and subsequently remove them from the image. However, coconut trees planted near the beach could be detected and discriminated from oil palm trees due to different backscattering effects from the surroundings. Figure 7a shows one of the locations of misclassified coconut as oil palm. We used the land use map produced by the Department of Agriculture, Peninsular Malaysia and Google Earth images to manually identify coconuts that were misclassified as oil palm (Figure 7c).

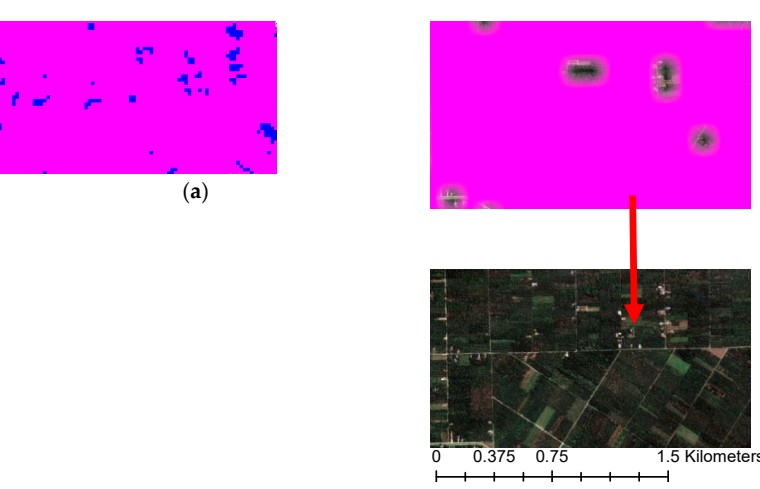

(**a**)

(**b**)

**Figure 7.** Misclassified coconut as oil palm: (**a**) Coconut land use (blue colored pixels) extracted from Land Use map overlaid on oil palm area classified in this study (pink); (**b**) Google Earth image showing coconuts.

Although we found our technique overestimated some oil palm areas, our technique generally identified oil palm well compared to the land use map produced by the Department of Agriculture, Peninsular Malaysia. For example, Figure 8a shows a huge area of oil palm that was classified as forest (dark green color surrounding the pink polygon (oil palm)) by the Department of Agriculture. However, Figure 8b (results of this study) correctly identified the area as oil palm. We cross checked the area with the Google Earth image and found that they were indeed oil palm areas (Figure 8c). This finding shows that the overestimated oil palm area in this study is justifiable.

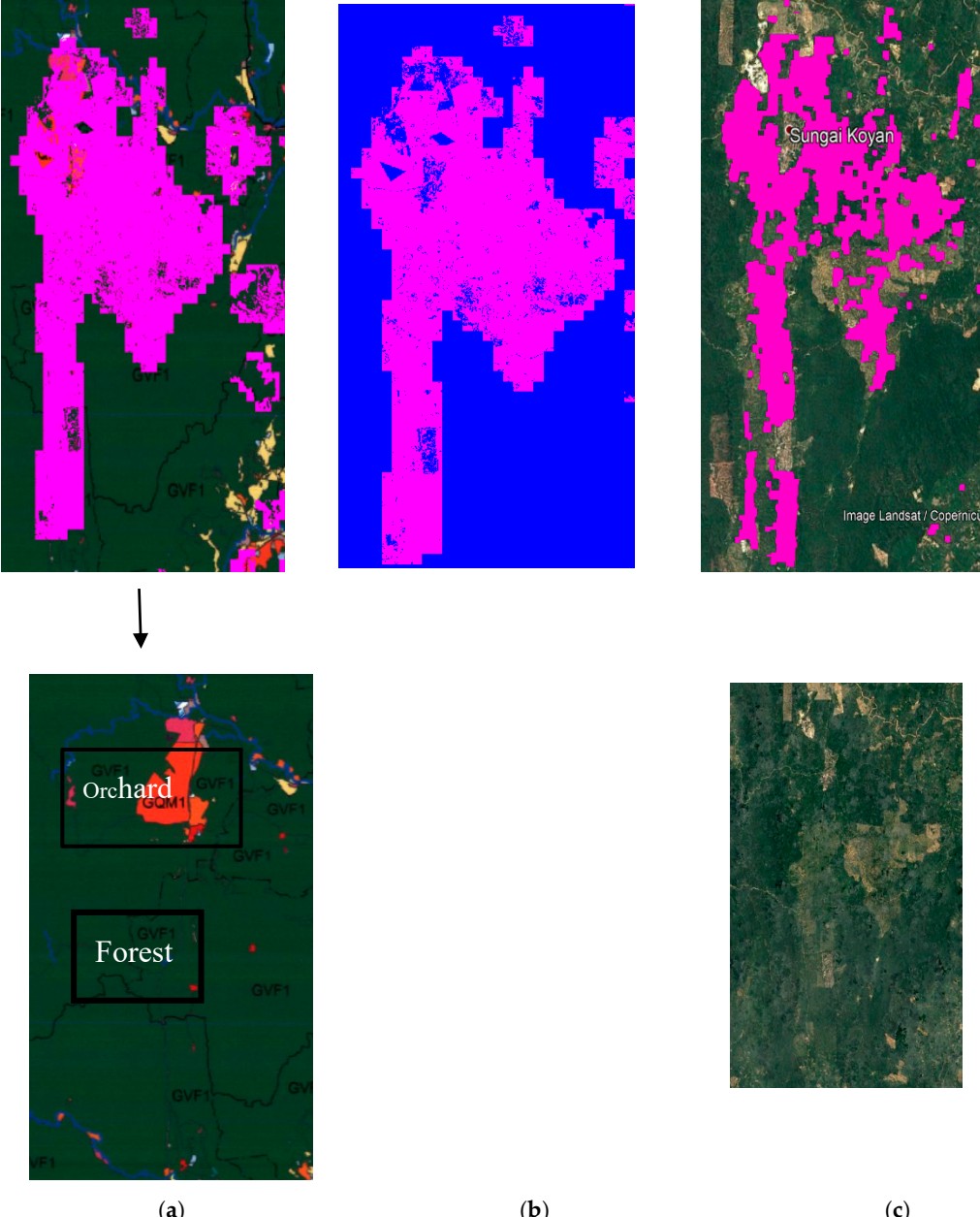

         (**a**)                                  (**b**)                                  (**c**)

**Figure 8.** (**a**) Oil palm that was detected as forest by the Department of Agriculture; (**b**) oil palm that was correctly detected as oil palm in this study; (**c**) Google Earth image showing that the area is indeed oil palm.

### 4.3. Forest Conservation and Sustainability

As far as agricultural land expansion is concerned in the tropics, the vast majority of agricultural growth has occurred in forested areas [67]. Oil palm plantations have always been criticized as being

responsible for the large-scale deforestation of tropical regions. Forested areas have been recklessly replaced with oil palms due to their high economic value. In Malaysia, approximately 650,000 ha of forest was found to have been replaced with oil palm in 2014 [68] and forest cover loss is occurring continuously [69]. Forests in the tropics have high conservation value (HCV) in terms of biological, ecological, social, or cultural values, and warrant protection because they are globally significant [70]. However, the expansion of oil palms in the tropical lowland areas has been found to be the major cause of biodiversity loss [71–73], amplification of global warming and climate change [74], and social conflict between oil palm growers and the indigenous people who depend on the forest for their livelihood [75]. Mapping and monitoring the extent of oil palm plantations is therefore essential for understanding the amount and magnitude of deforestation, its environmental consequences, informing policy decisions, and planning forest resources.

Maps showing the spatial and temporal (through multi temporal data analysis) distribution of oil palm plantation and other land use changes can assist in identifying illegal land conversion, such as if forests were converted into oil palm plantations. This is the practice adopted by the Roundtable Sustainability of Palm Oil (RSPO), a non-governmental organization that utilizes remote sensing data to map various land use and land cover changes to trace any intrusion of oil palm plantation into forest or other HCV areas such as peat land, and local people's land. This practice is vital to ensuring the sustainable development of the oil palm industry to prioritize conservation and protection of natural resources and biodiversity [76]. A global analysis considering 20 countries reveals that oil palm cultivation is associated with deforestation [73]. In Southeast Asia, they reported almost 50% of the sampled plantations were originally forest in 1989. A similar trend was also noticed in South America (31%), Mesoamerica (2%), and Africa (7%). In Kalimantan, approximately 90% of the forested areas were replaced by oil palm and mapping these areas has helped identify non-deforested locations that can be protected from further conversion [77]. Another HCV area often considered for planting oil palm is peat land, which contributes significantly to the emission of GHGs to the atmosphere when disturbed [74]. Mapping and demarcating the boundaries of oil palm plantations can aid in detecting any unauthorized invasion to these lands. A previous study [78] discovered approximately half of the peat land in Sarawak had been transformed into oil palms. A regular monitoring, mapping, and reporting of the oil palm plantations in this region will enable timely actions to be taken to protect remaining peat lands.

The detection of oil palm boundaries and monitoring their growth over time is also important to assess the impact of deforestation on biodiversity loss. Oil palm plantations generally consist of homogeneous tree species and thus they do not support the existence of many fauna or flora [79,80]. Expanding oil palm plantation over forests is a major threat to large fauna such as elephant, rhino, tiger, etc. in Malaysia [72]. Malaysia has the highest number of International Union for Conservation of Nature (IUCN) Red List of threatened species per square kilometer (approximately 0.03) relative to palm oil production [71]. Conversion of peat swamp forest to oil palm has resulted in the destruction of many bird species in Peninsular Malaysia, Borneo, and Sumatera [17]. Vijay et al [73] revealed the impact of oil palm plantation on recent deforestation and biodiversity loss in 20 countries using high resolution satellite images. They examined the recent growth in oil palm areal extent and modeled future expansions to propose conservation prioritization for threatened and small-range species of birds and mammals in areas that are vulnerable to future oil palm plantation. They recommended areas in the Amazon, Brazilian Atlantic Forest, Liberia, Cameroon, Malaysia, and western Indonesia for a combination of small-range and threatened mammal species. Using maps of recent conversions and remaining forests, Abram et al [81] analyzed the spatial and economic components of forest conversion to oil palm within a tropical floodplain in the Lower Kinabatangan, Sabah, Malaysian Borneo. They produced maps of oil palm and forests that assisted the discovery of a large area of unprotected forest rich in biodiversity and connectivity (64%) allocated for future oil palm development. Such studies are crucial to guide policies to reduce deforestation as a consequence of oil palm plantation and to protect ecosystems that are rich in biodiversity.

## 5. Conclusions

Classifying oil palm and knowing its geographical distribution is essential to demarcating the oil palm from other nearby land covers such as forest, other vegetation, buildings, and bare land, as well as for the accurate estimation of oil palm area coverage. The accurate classification of areas covered by oil palm trees is important for many purposes, including estimation of oil palm yield and biomass per hectare, automatic tree counting, change detection, and age estimation. The improved oil palm classification can aid in the monitoring of oil palm area expansion to avoid further loss of forest and biodiversity, especially on land with high conservation value. The application of remote sensing helps in this regard to acquire valuable and otherwise expensive information. This study demonstrates that the exploitation of multi sensor data; ALOS PALSAR, Sentinel, Landsat, and images from Google Earth are very effective for oil palm classification and can be advanced into a plantation monitoring system for sustainable management of oil palm and other natural resources. The HH, HV, and manipulated polarizations of ALOS PALSAR data can discriminate oil palm trees from forest and rubber trees, whereas HH polarization can identify and discriminate misclassified water bodies as oil palms. The red spectra L-band from Landsat data could identify built up areas and VH polarization from Sentinel C-band data and is good in detecting bare land that tend to be misclassified as young oil palm. For future studies, it is suggested that the removal of coconut plantations can be carried out in a better way using higher resolution data.

**Author Contributions:** N.E.M.N. carried out the field data collection, data processing, analysis, and writes the original draft. K.D.K. designs the conceptualization, supervised, analyse data and write the original draft. A.P.C. and L.Y. were supervise and editing the manuscript. All authors have read and agreed to the published version of the manuscript.

**Funding:** The authors extend their thanks to Universiti Teknologi Malaysia, the Ministry of Education, Malaysia and WNI WXBUNKA Foundation, Japan via the UTM High Impact Research grant (Q.J130000.2452.08G51), Fundamental Research Grant (R.J130000.7852.5F216) and research grant R.J130000.7352.4B406 respectively for providing research funding.

**Acknowledgments:** Our profound gratitude goes to USGS department for providing free Landsat images data. We would also like to thank Alaska Satellite Facility (ASF) for providing free SENTINEL data. We acknowledge Department of Agriculture and Malaysian Palm Oil Board for providing land use data and oil palm area statistics to conduct the study.

**Conflicts of Interest:** The authors declare no conflict of interest.

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
