# Peer review of "Synergy of Active and Passive Remote Sensing Data for Effective Mapping of Oil Palm Plantation in Malaysia"

_forests, doi:10.3390/f11080858_

Round 1
Reviewer 1 Report
The authors did a lot of work and several changes in the methodology, results, and discussion. The manuscript definitively improved and it is much better than the previous version. However, the written still needs some improvements. I suggest to the authors to try to connect the terms used in Figure 2 in the methodology and separate the topic methodology in subtopics. For instance, it is better to explain the pre-processing from optical and SAR separately and with more details connect this with the flow-chart of Figure 2. Although the authors sent the paper for English review, it is still noticeable a problem of style. Please see some examples below. I would suggest to the authors to review the Methodology, Results, and Discussion items specifically carefully again.
In the methods section, you mention that:
- “However, it should be noted that when extracting texture information, the speckle can become a valuable source containing rich texture information or can destroy some of the textures”.
I would not use the word destroy, I would say can cause loss of information since make it smoother
- This needs to be rewritten: “ In this study, the trial and error method adopted showed that without applying the Lee filter an overall classification accuracy of ~50% was obtained, but after applying the filter the accuracy increased to >70% mainly because the speckle filter reduced errors in the data and the texture measure (GLCM) further enhanced the backscatter signals originating particularly from oil palm plantations. “
Why trial? Maybe replace by “in this study the framework and evaluation of the error” …..
Again this is not the best way to write it: “ mainly because the speckle filter reduced errors in the data”.
The speckle filter reduce the salt and pepper effect from the SAR images.
- In : “ The HH, HV, HH-HV and HH/HV images from the ALOS PALSAR, VH from Sentinel-1 and Red band from Landsat data were used to identify backscattering (radar) and reflectance (optical) threshold values that can differentiate oil palm from other land covers/uses.”
How? By using a classification method, algorithm? Could you also include some a reference of the methods here?
- Here “First, several ROIs (Figure 3) were extracted for each land cover (oil palm, forest, bare land, water and built up areas) classes”. I would write “(Figure3 and Table 2)”.
- This also needs review: “ This is rather a tedious process that requires careful selection of pure pixels representing each land cover class in the study region.”. Maybe tedious is not the best word here.
Reviewer 2 Report
Line 129: Table 1. Please change the Sentinel resolution/pixel spacing
accordingly with the mode you used in this research.
https://sentinel.esa.int/web/sentinel/user-guides/sentinel-1-sar/resolutions/level-1-ground-range-detected
Line 420: Please re-write the sentence. The authors should not make a
statement if the data is not shown.
Round 2
Reviewer 1 Report
Thank you to the authors for the effort. The paper is much better now.
This manuscript is a resubmission of an earlier submission. The following is a list of the peer review reports and author responses from that submission.
Round 1
Reviewer 1 Report
I believe the authors present research that is relevant to forest conservation However, although the authors argue that the research has relevance to conservation and sustainability studies, the topic is not revisited in the discussion and modestly revisited in the conclusions. The result is the presentation that is methods/process paper that might be more appropriate for a remote sensing journal, though I am sure it will be of interest to some readers of Forests.
Reviewer 2 Report
Dear authors,
The paper combines active and passive remote sensing data for the effective mapping of oil palm plantation in Malaysia. The paper is interesting and the classification shows high overall accuracy and kappa. The study is interesting but not original since we found in the literature many studies using different approaches combining SAR and Optical datasets to palm oil plantation classification. The importance of this work needs to be clarified and highlighted in the manuscript.
I am concerned about some points:
The introduction needs to be improved including explicitly the importance and contribution of this work and the English language needs some review.
The classification method and threshold need to be more detailed. It is not clear how the thresholds were extracted. The variables of textures generated from SNAP from SENTINEL 1 and ALOS 2 needs to be described and detailed about how they were evaluated needs to be provided.
The validation section needs to be moved to the results.
The discussion section is a mix of the results of validation and discussion. I would advise the authors to separate well those 2 sections (Results and Discussion).
Line 51: Delete “Satellite borne” starts the sentence in Remote sensing…
Line 56: What would be conventional?
Line 60 and 61: Does the optical data tend to underestimate the are of oil palm plantation only due to the cloud cover problem? What about the shadows, saturation of the signal? Please clarify it.
Lines from 63 to 65 needs to be rewritten since it is a bit confused, also it needs a reference by the end of the sentence.
Lines 65: The microwave from the RADAR has capabilities to penetrate the vegetation and it is sensitive to the geometry of the target in this case forest structure. It is important to leave this clear.
Lines from 113 to 120: Include the full description of the SAR datasets used. The ALOS datasets used was SLC or they were from JAXA MOSAIC backscattering datasets? Sentinel 1 were SLC or GRD? Please give all information.
Figure 2: I do not understand the box Land Use Map and GEE. This is an input for the validation or a output from the validation?
Line 150: It was no clarity about how you found the thresholds. Did you use histograms? Signatures? Please explain it in details.
Line 157. Why did you use HH and Lee filter? Please explain and cite some references on your approach.
Line 180. The textures (GLCM) was extracted only for Sentinel 1? What about ALOS?
The item 4.1 on Validation of Oil Palm mapping should be part of the results section.
Reviewer 3 Report
This manuscript is another case study on SAR and optical data which would be of interesting for research community. However, it requires modifications and justification to improve the quality of the manuscript and its outreach to readers. Would kindly ask the authors to consider the following comments for a potential improvement. Also, would suggest to improve the methods section.
1. Introduction:
- Line 43 - Do you mean front or fond?
- Lines 63-64: The usefulness of radar remote sensing to classify oil palm plantation is mentioned. It would be interesting to add some references to this statement.
- Line 68: Please clarify reference [18]. Authors explained the advantage of radar in previous lines (66-68). However, the reference added in the manuscript used WorldView-2.
- Lines 77-79: This statement should be added into the discussion section with a comparative between their findings and the authors findings.
- Lines 82-83: Duplicated references. Please fix.
2. Materials and Methods:
2.2. Datasets:
- Line 117: It would be better if author can provide the GloVis GloVis website here.
- Lines 122-123: Please provide a reference (scientific journal if possible) of the Land Use map used.
- Line 128: I would advise the authors to re-structured Table 1. Keep the data (ALOS PALSAR 2, Landsat-8 OLI, Sentinel-1) and add some characteristics of the satellites such as beam mode, orbit, incidence angle, etc., rather than use a website. This will help readers going through the manuscript.
2.3 Methods:
- Lines 141-142: It would be better if author can provide a reference to Lee filter.
- Lines 144-146: It is mentioned that a speckle filter was applied prior to the extraction of GLCM textures. However, when extracting texture information, the speckle can become a valuable source containing rich texture information or can destroy some of the textures. Have a quantitatively test carried out to check to what extent speckle filtering affects texture?. Which were the windows size and the orientations chosen to extract the GLCM textures? Also, please add a reference to GLCM.
- Lines 147-148: Please rewrite this paragraph to improve clarity.
- Lines 148-150: Please rewrite the sentence 'The processed ALOS PALSAR 2 images were manipulated by finding the difference (HH minus HV) and ratio (HH over HV) between both HH and HV ALOS PALSAR 2 polarization'.
- Line 167: A reference to Sentinel is provided when authors are referring to Landsat. Please modify.
- Lines 169-172: It is not clear to me why these two sentences are in the methods section. If it is the case should be in discussion or conclusions.
- Line 175: None of Sentinel-1 Level 1 products has a 20m spatial resolution. Please change 20m to 10m.
- Line 180-181: Same as comment above, I think GLCM textures should be checked with and without applying speckle filter. Please add reference and comment the windows size and orientations chosen when Sentinel-1 was used.
- Lines 189-193: It is previously mentioned that the land use map of Peninsular Malaysia was used '...to visually inspect and compare with the oil palm areas classified in this study'. Why the land use map wasn't used to extract the ground truth polygons? An image segmentation of the map could have been carried out to extract the polygons.
3. Results:
- Lines 205-208: Authors should not make any statement based on results that are not shown. If this statement is based on the previous references then I would suggest authors to rewrite the statement if not, then should be remove it.
- Line 209: Modify "polaization" by "polarization".
- Line 230: The Analysis of Variance is mentioned. However, it has not been introduced in the methods section.
- Lines 258-261: The sentence seems to explain the value of C-band satellites to distinguish pure bare land and bare land because of the C-band canopy penetration ability, and the reference added to point this out used X band. Would you please clarify it or modify the reference.
- Line 281: Figure 4(b) and (e) - Are the Y-axis values really in dB?
- Line 284: Change "Sentinel 2" by "Sentinel-1".
- Line 306: Figure 5 - Improve resolution of all maps.
4. Discussion:
- Line 337: Table 5 - Latest study is not reference [30].
- Line 340: Although is well-known in remote sensing, confusion matrix and accuracy assessment metrics should be explained in methods section.
- Line 352: Table 9 - Please review reference.
- Line 358: Reference [56] is an interview. Please use scientific journals and books peer review.
